# The Health Impacts of Better Access to Axicabtagene Ciloleucel: The Case of Spain

**DOI:** 10.3390/cancers16152712

**Published:** 2024-07-30

**Authors:** Raúl Córdoba, Lucía López-Corral, María Presa, Victoria Martín-Escudero, Sachin Vadgama, Miguel Ángel Casado, Carlos Pardo

**Affiliations:** 1Lymphoma Unit, Department of Haematology, Fundación Jiménez Díaz University Hospital, 28040 Madrid, Spain; raul.cordoba@fjd.es; 2Department of Haematology, Hospital Universitario de Salamanca, Instituto de Investigación Biomédica de Salamanca (IBSAL), Centro de Investigación Biomédica en Red Cáncer (CIBERONC), Centro de Investigación del Cáncer-IBMCC (USAL-CSIC), 37007 Salamanca, Spain; lucialopezcorral@usal.es; 3Health Economics Department, Pharmacoeconomics & Outcomes Research Iberia (PORIB), 28224 Madrid, Spain; ma_casado@porib.com; 4Market Access, Reimbursement & Health Economics and Outcomes Research Department, Gilead Sciences, 28033 Madrid, Spain; victoria.martin-escudero@gilead.com (V.M.-E.);; 5Kite, a Gilead Company, Uxbridge UB11 1BD, UK; sachin.vadgama@gilead.com

**Keywords:** axicabtagene ciloleucel, diffuse large B-cell lymphoma, primary mediastinal B-cell lymphoma, value, health outcomes, advanced therapies report

## Abstract

**Simple Summary:**

Axicabtagene ciloleucel (axi-cel) has been shown to improve the health outcomes of patients with relapsed/refractory (R/R) diffuse large B-cell lymphoma (DLBCL); however, the actual number of patients treated in Spain is lower than the epidemiology estimations. The aim of our study was to assess the value of axi-cel versus chemotherapy in patients with R/R DLBCL after ≥2 lines of therapy based on the number of patients treated. Considering that the entire cohort was eligible for treatment with axi-cel (n = 490) compared to the currently treated population (n = 187), the use of axi-cel rather than chemotherapy in all eligible patients could lead to 2173 life years gained and 1706 quality-adjusted life years. Furthermore, if all eligible patients were treated with CAR T-cell therapy, an additional 85 patients would be alive, and 78 patients would be alive without disease progression at 5 years.

**Abstract:**

In this study, the health impacts of improving access to treatment with axicabtagene ciloleucel (axi-cel) was assessed in patients with relapsed/refractory diffuse large B-cell lymphoma after ≥2 lines of therapy in Spain. A partitioned survival mixture cure model was used to estimate the lifetime accumulated life years gained (LYG) and quality-adjusted life years (QALYs) per patient treated with axi-cel versus chemotherapy. Efficacy data were extracted from the ZUMA-1 trial for axi-cel and from the SCHOLAR-1 study for chemotherapy. In the base case, the incremental outcomes of axi-cel versus chemotherapy were evaluated in a cohort of 187 patients treated with CAR T-cell therapies, as reported by the “Spanish National Health System Plan for Advanced Therapies”, and in the alternative scenario in the full eligible population based on epidemiological estimates (n = 490). Taking those currently treated with axi-cel, compared with chemotherapy, axi-cel provided an additional 1341 LYGs and 1053 QALYs. However, when all eligible patients (n = 490) were treated, axi-cel provided an additional 3515 LYs and 2759 QALYs. Therefore, if all eligible patients were treated with axi-cel rather than those currently treated as per the registry (n = 187), there would have been an additional 303 patients treated, resulting in an additional 2173 LYGs and 1706 QALYs in total. The lack of access in Spain has led to a loss of a substantial number of LYGs and QALYs, and efforts should be made to improve access for all eligible patients.

## 1. Introduction

Non-Hodgkin lymphoma (NHL) is the most common haematologic malignancy [1]. In 2020, 544,352 new NHL cases were diagnosed worldwide, representing an age-standardized incidence rate of 5.8/100,000 patients per year [2].

Diffuse large B-cell lymphoma (DLBCL), the most common type of malignant lymphoma worldwide, constitutes 30–58% [3] of NHL malignancies and 80−90% of all aggressive (high-grade) lymphomas [4]. In Spain, the RELINF project, created by the Spanish Lymphoma and Autologous Bone Marrow Transplant Group, was designed as an online platform for the prospective registry of data on newly diagnosed cases of lymphoid neoplasms in Spain [5]. From January 2014 to July 2018, a total of 828 patients with DLBCL (9.1% of B-NHL patients) were registered [5].

It is estimated that approximately 30−40% of DLBCL patients will relapse, and 10% will be refractory to first-line therapy. The prognosis for patients who are resistant to primary therapy or who experience multiple relapses is considerably worse, with an estimated median overall survival of three months for those who relapse from high-dose therapy and autologous stem-cell transplantation (auto-SCT) [6]. To characterize the response rates and survival of patients with refractory disease, a patient-level pooled retrospective study among patients with relapsed/refractory (R/R) large B-cell lymphoma (LBCL) treated with conventional chemotherapy was developed [7]. In SCHOLAR-1, patient-level data were extracted from two observational cohorts and two phase III randomized controlled trials, showing a pooled median overall survival (OS) of 6.3 months [7].

Until the availability of chimeric antigen receptor (CAR) T-cell therapies, most patients with R/R DLBCL had limited treatment options because the alternatives for patients who were ineligible for auto-SCT were limited mainly to chemotherapy regimens used with palliative intent, given their limited efficacy [8,9,10,11]. CAR T-cell therapies have revolutionized the treatment of LBCL, leading to unprecedented improvements in OS and response rates [12,13,14]. Axicabtagene ciloleucel (axi-cel) was approved by the European Medicines Agency (EMA) in 2018 for the treatment of adult patients with R/R DLBCL and primary mediastinal large B-cell lymphoma (PMBCL) after two or more lines of systemic therapy [15]. Axi-cel is an autologous anti-CD19 CAR T-cell therapy and binds to CD19-expressing target cells, the CD28 and CD3-zeta domains that activate downstream signalling cascades, leading to T-cell activation [15]. The efficacy of axi-cel was demonstrated in the pivotal ZUMA-1 study, a multicentre, phase II trial in which 111 patients with refractory DLBCL, PMBCL or transformed follicular lymphoma were enrolled [13].

Comparing the 2-year outcomes of ZUMA-1 with those of SCHOLAR-1, axi-cel demonstrated higher objective response rates (83% with axi-cel vs. 34% with chemotherapy) and a 73% reduction in mortality risk compared to salvage chemotherapy [16].

After 5 years of follow-up of ZUMA-1 (median 63.1 months), the median duration of complete response (achieved by 58% of patients) with axi-cel was 62.2 months. The median progression-free survival (PFS) was 5.9 months, with an estimated 5-year PFS rate of 31.8%, and the median OS was 25.8 months, with a 5-year OS rate of 42.6% [17]. Among those patients who achieved CR, the median OS was not reached, and the 5-year OS rate was 64.4% [17]. 

A cost-effectiveness analysis comparing axi-cel vs. salvage chemotherapy in patients with relapsed/refractory DLBCL from the Spanish payer perspective has been published previously, positioning axi-cel as a potentially cost-effective option [18]. Axi-cel has also been shown to be cost-effective against another CAR T-cell therapy (tisagenlecleucel), resulting in an incremental cost–utility ratio of EUR 13,049 per quality-adjusted life year (QALY) [19]. This value stands significantly below the frequently used willingness-to-pay thresholds of EUR 22,000/QALY and EUR 60,000/QALY in Spain [20,21].

In Spain, CAR T-cell therapies were included in the public health provision in 2019. To regulate and monitor use, the Spanish Health Ministry developed the “Spanish National Health System Plan for Advanced Therapies”, a tool to organize the use of CAR T-cell therapies in an equitable, safe and efficient manner to guarantee quality, safety and efficacy standards. The plan rules that every patient eligible for CAR T-cell therapy should receive prior approval from the Ministry of Health [22]. To date, the Ministry of Health has published four reports providing real-world data on the efficacy and safety outcomes of CAR-T-cell therapies in patients treated in Spanish hospitals [22]. However, according to the latest report published on 15 July 2022, the number of patients with R/R DLBCL treated with CAR T-cell therapies is lower than the epidemiological estimates of patients eligible for treatment.

The objective of this study was to assess the clinical value of better access to axi-cel in patients with R/R DLBCL who previously received two or more lines of treatment in Spain. We did this by comparing outcomes with the actual number of patients treated reported in the “Spanish National Health System Plan for Advanced Therapies” to the outcomes that would be attained if the full number of patients estimated to be eligible for CAR T-cell therapy for this indication were treated, based on epidemiological estimates.

## 2. Materials and Methods

### 2.1. Modelling Approach

A partitioned survival model (PSM) was adapted to evaluate the health outcomes yielded by axi-cel compared to those yielded by rituximab-based salvage chemotherapy for adult patients with R/R DLBCL after ≥2 lines of treatment. Health outcomes were expressed as life years gained (LYG), quality-adjusted life years (QALY) gained and the number and percentage of patients alive and without progression at 6 months and at 1, 2, 5 and 10 years.

According to previous cost-effectiveness analyses (CEAs) evaluating CAR T-cell therapies in LBCL R/R [23,24,25,26], the PSM included preprogression, postprogression and death health states (Figure 1), with patients transitioning from one state to another based on OS and PFS curves.

Due to the potentially curative nature of axi-cel, a mixture cure model was used to represent the axi-cel OS curve, an approach widely used in prior CEAs [23,24,25,26]. With this approach, a proportion of individuals are expected to achieve long-term survival, with mortality rates equivalent to those of the general population.

All patients were subjected to background noncancer-related mortality analysis, which was derived from age- and sex-matched mortality data from population life tables [27].

To avoid overly penalizing interventions that generate most of their benefits in the future, a discount rate for health outcomes was not considered. To accurately capture disease progression and health outcomes, a monthly cycle was used, and a half-cycle correction was applied [28].

All the assumptions and parameters included in the analysis were reviewed and validated by two haemato-oncology experts.

### 2.2. Population

The patient cohort considered in the simulation included adult patients with R/R DLBCL after ≥2 systemic therapies, in alignment with the population included in the ZUMA-1 trial [8] and the marketing authorization for axi-cel [15].

In the base case, the health outcomes associated with axi-cel versus chemotherapy were evaluated in a cohort of 187 patients, the number of patients treated with CAR T-cell therapies in one year according to the “Spanish National Health System Plan for Advanced Therapies” reports published between June 2021 and July 2022 [22]. In an alternative scenario, the results were analysed in a cohort of 490 patients according to the epidemiological estimate of the number of patients eligible for treatment with CAR T-cell therapy in Spain made by the Spanish Society of Hospital Pharmacy [29].

Patient characteristics were assumed to be in line with those reported in the ZUMA-1 trial for the modified intention-to-treat population (patients who received infusion of axi-cel): 58 years old and 33% women [13].

### 2.3. Comparator

Currently, in Spain, there is no clearly defined standard-of-care therapy for the treatment of R/R DLBCL with more than two previous treatments. In most cases, patients are treated with rescue chemotherapy. The following pools of chemotherapy regimens were selected as comparators for analysis based on common clinical practice in Spain: R-GEMOX (rituximab, gemcitabine and oxaliplatin) and R-ICE (rituximab, ifosfamide, carboplatin and etoposide).

### 2.4. Clinical Data

For axi-cel, OS and PFS data were obtained from the ZUMA-1 clinical trial after four years of follow-up [30]. In the absence of a comparator arm in ZUMA-1, comparative data were obtained from the SCHOLAR-1 study, which has been used extensively in both Health Technology Assessment and regulatory studies to evaluate the effectiveness of axi-cel [16,31,32].

For each survival curve, six parametric distributions were generated to extrapolate the OS and PFS curves beyond the study observation periods, selecting the distribution based on statistical goodness-of-fit (using the Akaike information criterion and Bayesian information criterion) and clinical validation.

Based on the ZUMA-1 four-year follow-up data [30], the mixture cure model stratified the population into cured and noncured groups, estimating a cure fraction of 41%. The final OS curve was derived from the curves of the two groups weighted by the cure fraction and extrapolated beyond the follow-up period by a lognormal parametric distribution. For the axi-cel PFS data, the best-fitting distribution was the Gompertz distribution (Figure 2a).

To estimate the OS associated with chemotherapy, propensity score matching was used, based on the National Institute for Health and Care Excellence recommendations [33], to adjust for differences between the baseline characteristics of the studies and to provide a population comparable to the population included in the ZUMA-1 trial. When the OS curve was extrapolated beyond the end of the SCHOLAR-1 study, a Gompertz distribution was found to be the best fit. SCHOLAR-1 was not used to assess progression, and therefore, PFS data were not available; therefore, the chemotherapy PFS curve was estimated by the axi-cel ratio between PFS and OS at each time point (Figure 2b).

### 2.5. Utilities

To estimate the number of QALYs gained, health utility values were associated with each health state. The utility value used in the base case for the preprogression state was obtained from the ZUMA-1 trial, specifically from the safety population, which assigned a utility value of 0.72 [34]. For the postprogression state, a utility value of 0.39 was considered, which was obtained from the literature [35].

According to the literature [36], patients who were free of disease progression during the first 36 months were assumed to have age-specific general population utility value, which was derived from mean EQ-5D-5L scores obtained from the Spain National Health Survey [37].

Patients treated with axi-cel were assigned a utility decrease of −0.05, which is associated with CAR T-cell-related toxicity, in the first model cycle [34]. The SCHOLAR-1 study did not reflect the chemotherapy toxicity profile; hence, the analysis did not consider the utility decrement associated with chemotherapy-related adverse events.

## 3. Results

### 3.1. Base Case

After extrapolating the Kaplan–Meier curves over a lifetime horizon, the mean OS modelled for axi-cel was 11.6 years versus 4.4 years for chemotherapy. Axi-cel increased the mean PFS by 6.9 years (11.0 years with axi-cel versus 4.1 years with chemotherapy).

Considering the survival benefits associated with axi-cel and considering that 187 patients were treated with CAR T-cell therapy, axi-cel generated an increase of 162% in LYGs (1341 LYGs) and 165% in QALYs (1053 QALYs) compared with chemotherapy (Table 1).

### 3.2. Alternative Scenario

In the alternative scenario, in which 490 patients could be treated with CAR T-cell therapy, the health outcomes of patients treated with axi-cel increased by 3515 LYGs and 2759 QALYs versus chemotherapy (Table 2).

### 3.3. Comparison between the Base Case and Alternative Scenario

Compared to the currently treated population (n = 187), an additional 303 patients would have been eligible for treatment with axi-cel; therefore, we estimate there would be an additional 2173 LYGs and 1706 QALYs generated in the Spanish population (Table 3).

Considering the modelled efficacy data, if all eligible patients were treated with CAR T-cell therapy, an additional 85 patients would be alive, and 78 patients would be alive without disease progression at 5 years (Figure 3).

## 4. Discussion

In recent years, CAR T-cell therapies have revolutionized the therapeutic landscape for R/R DLBCL patients in terms of response and survival [7]. The Spanish Ministry of Health has developed the “Spanish National Health System Plan for Advanced Therapies” to ensure the equitable, efficient and safe use of these treatments [13]. In addition, it enables the collection of real-world data from patients treated with CAR T-cells (13). The “Spanish National Health System Plan for Advanced Therapies” reports clearly show a gap between the number of patients treated with CAR T-cell therapies and the estimated number of eligible patients in Spain.

To our knowledge, this is the first study to estimate the impact of inefficient access to CAR T-cell therapies on the health of the Spanish population in Spain. This analysis presents an estimation of the health outcomes that could be gained for this patient population, which faces poor prognosis in the absence of CAR T-cells. This study provides additional evidence in support of treatment with CAR T-cell therapies by highlighting the improvements in survival and quality of life that could be achieved with better patient access to treatment. At present, there are several barriers responsible for delaying the use of CAR T-cell therapies in Spain, including the complexity of the multistep administrative application process to the Ministry of Health, the late referral of high-risk patients and the distance from the referring centre to the qualified infusing centres, among other factors [38]. Potential actions to address these issues include streamlining the bureaucratic process and increasing coordination between the Ministry of Health and Autonomous Communities, simplifying data reporting for healthcare professionals, reinforcing continuing medical education, and providing greater financial support for patients who must travel to a qualified centre. Other plausible strategies that could also improve the performance of the process may include patient-reported outcome measurements (PROMs) and patient-reported experience measurements (PREMs), not only in clinical trials but also when collecting real-world data in clinical practice.

There are some limitations intrinsic to this study that need to be mentioned. First, the patient cohorts considered in the analysis were modelled according to the available published data. The number of patients treated with CAR T-cell therapy (n = 187) [22] could have been underestimated, as the registry of some patients in the database might be delayed and the data may not have been correctly updated. In addition, for simplicity, we assumed that the total number of patients treated with CAR T-cell therapy in Spain would be treated with axi-cel, without taking into account other CAR T-cell therapy. In addition, the estimated number of patients eligible for treatment with CAR T-cell therapy (n = 490) [29] might not be precise since it is based on prevalence figures from previous years. More recent and updated epidemiologic studies might enable more accurate estimation.

Second, since ZUMA-1 is a single-arm trial that lacks a comparator, comparative effectiveness between axi-cel and chemotherapy was assessed through a synthetic control study (SCHOLAR-1). This may be associated with a potential bias since there may be differences in patient characteristics between the two studies, such as the number of previous treatments or the proportion of patients undergoing stem cell transplantation [26,39]. In addition, the chemotherapy regimens defined in this study, which are most frequently utilized in clinical practice in Spain, do not fully coincide with those described in SCHOLAR-1, possibly leading to a misfitting of the survival curves.

Another limitation of this study is the possible marketing authorization of new treatments, as well as the use of axi-cel in the second-line treatment of DLBCL, which could reduce the number of patients eligible for CAR T-cell therapies.

The results obtained in the present study highlight the effectiveness of axi-cel compared to chemotherapy for R/R DLBCL treatment after two or more lines of systemic therapy. The findings obtained represent an important tool that provides useful information for healthcare decision-makers to find ways to improve the patient access to CAR T-cell therapy. In this sense, special emphasis is placed on the main difficulties that the Spanish National Health System is currently facing and the plausible short-term strategies that could be implemented to ultimately improve survival and quality of life for patients and to diminish the burden of the disease.

## 5. Conclusions

This work focuses on the potential health benefits that could be achieved for R/R DLBCL patients previously treated with two or more lines of systemic therapy when access in Spain is improved, as it is currently suboptimal. According to the results obtained in the present analysis, treating all eligible R/R DLBCL patients with axi-cel versus chemotherapy with respect to currently treated patients could lead to increases of 2173 LYGs and 1706 QALYs, respectively, and a 162% increase in the number of patients who are alive (n = 85) and alive without disease progression (n = 78) at 5 years.

These results highlight the need to implement plausible short-term strategies that improve access to CAR T-cell therapies, improve the survival and quality of life of patients with R/R DLBCL and diminish the burden of disease.

## Figures and Tables

**Figure 1 cancers-16-02712-f001:**
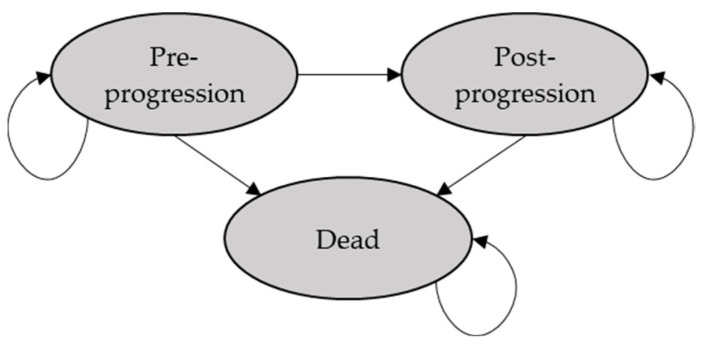
Diagram of the analytic decision model. **Notes:** The PSM allows for the estimation of health state occupancy through the OS and PFS curves. For each therapy (axi-cel or chemotherapy), patients in the model would transition into the preprogression health state (survival and no progress), postprogression health state (survival but progress) and death health state. The patients enter the model from the preprogression health state. When the patients transition to a postprogression health state, they can only maintain in this state or die.

**Figure 2 cancers-16-02712-f002:**
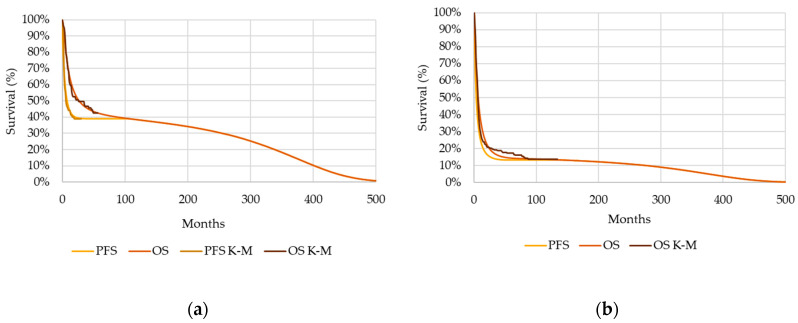
(**a**) OS and PFS curves for patients treated with axicabtagene ciloleucel and (**b**) chemotherapy. **Notes:** A specific parametric distribution was selected to extrapolate the OS and PFS Kaplan–Meier curves beyond the study observation periods over a lifetime horizon (log-normal for the OS of patients treated with axi-cel, Gompertz for the PFS of patients treated with axi-cel, and Gompertz for the OS of patients treated with chemotherapy). Abbreviations: K–M, Kaplan–Meier; OS, overall survival; PFS, progression-free survival.

**Figure 3 cancers-16-02712-f003:**
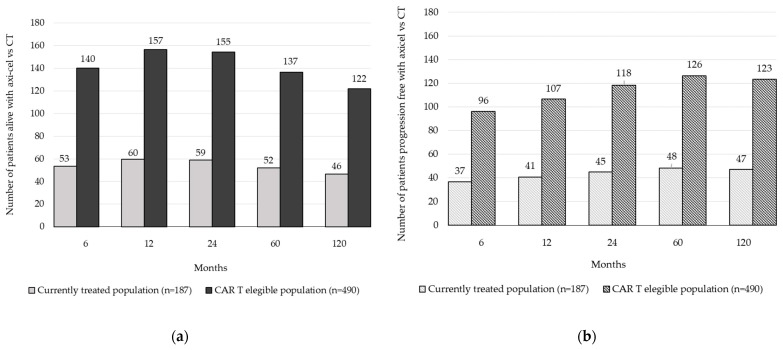
Incremental number of patients treated with axi-cel versus chemotherapy: (**a**) Incremental number of patients alive with axi-cel versus chemotherapy considering the currently treated population (187 patients) and the CAR T-cell-eligible population (490 patients). (**b**) Number of patients alive without disease progression with axi-cel versus chemotherapy considering the currently treated population (187 patients) and the CAR T-cell-eligible population (490 patients). Abbreviations: OS, overall survival; PFS, progression-free survival.

**Table 1 cancers-16-02712-t001:** Results obtained in the base case for a cohort of 187 patients (currently treated population).

Health Outcomes	Axi-Cel	Chemotherapy	Incremental
Total LYG per patient	2168	827	1341
LYGs in preprogression	2052	768	1284
LYGs in postprogression	116	59	58
Total QALYs per patient	1690	638	1053
QALYs in preprogression	1645 *	615	1030
QALYs in postprogression	45	23	22
Patients in preprogression state, n (%)			
Patients in preprogression at 6 months	102 (55%)	65 (35%)	37 (56%) **
Patients in preprogression at 1 year	82 (44%)	42 (22%)	41 (98%) **
Patients in preprogression at 2 years	74 (40%)	29 (15%)	45 (157%) **
Patients in preprogression at 5 years	73 (39%)	25 (13%)	48 (195%) **
Patients in preprogression at 10 years	72 (38%)	25 (13%)	47 (190%) **
Patients alive, n (%)			
Patients alive at 6 months	149 (80%)	95 (51%)	53 (56%) **
Patients alive at 1 year	121 (65%)	61 (33%)	60 (98%) **
Patients alive at 2 years	97 (52%)	38 (20%)	59 (157%) **
Patients alive at 5 years	79 (42%)	27 (14%)	52 (194%) **
Patients alive at 10 years	72 (38%)	25 (14%)	46 (184%) **

* QALYs in preprogression comprise total QALYs gained and the decrements due to AEs. ** Percentage increase in the number of patients in the preprogression state with axi-cel treatment versus chemotherapy at each time point. QALYs, quality-adjusted life years; LYG, life years gained.

**Table 2 cancers-16-02712-t002:** Results obtained in the alternative scenario for a cohort of 490 patients (CAR T-cell-eligible population).

Health Outcomes	Axi-Cel	Chemotherapy	Incremental
Total LYG per patient	5681	2166	3515
LYGs in preprogression	5377	2013	3363
LYGs in postprogression	304	153	151
Total QALYs per patient	4430	1671	2759
QALYs in preprogression	4311 *	1611	2700
QALYs in postprogression	119	60	59
Patients in preprogression state, n (%)			
Patients in preprogression at 6 months	268 (55%)	172 (35%)	96 (56%) **
Patients in preprogression at 1 year	216 (44%)	109 (22%)	107 (98%) **
Patients in preprogression at 2 years	194 (40%)	76 (15%)	118 (157%) **
Patients in preprogression at 5 years	191 (39%)	65 (13%)	126 (195%) **
Patients in preprogression at 10 years	188 (38%)	65 (13%)	123 (190%) **
Patients alive, n (%)			
Patients alive at 6 months	390 (80%)	250 (51%)	140 (56%) **
Patients alive at 1 year	317 (65%)	160 (33%)	157 (98%) **
Patients alive at 2 years	253 (52%)	99 (20%)	155 (157%) **
Patients alive at 5 years	207 (42%)	70 (14%)	137 (194%) **
Patients alive at 10 years	188 (38%)	66 (14%)	122 (184%) **

* QALYs in preprogression comprise total QALYs gained and the decrements due to AEs. ** Percentage increase in the number of patients in the preprogression state with axi-cel treatment versus chemotherapy at each time point. QALY, quality-adjusted life years; LYG, life years gained.

**Table 3 cancers-16-02712-t003:** Comparison of LYG and QALYs between the currently treated population (n = 187) and the CAR T-cell-eligible population (n = 490).

Health Outcomes	CAR T-Eligible Population * (n = 490)	Currently Treated Population * (n = 187)	Incremental (CAR T-Eligible Population vs. Currently Treated Population, n = 303)
Total LYG per patient	3515	1341	2173
LYGs in preprogression	3363	1284	2080
LYGs in postprogression	151	58	93
Total QALYs per patient	2759	1053	1706
QALYs in preprogression	2700	1030	1669
QALYs in postprogression	59	22	36

* Results obtained comparing axi-cel versus chemotherapy. CAR T, chimeric antigen receptor T cell; QALYs, quality-adjusted life years; LYG, life years gained.

## Data Availability

The original contributions presented in the study are included in the article, further inquiries can be directed to the corresponding author.

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
