# Peer review of "The Health Impacts of Better Access to Axicabtagene Ciloleucel: The Case of Spain"

_cancers, 2024, doi:10.3390/cancers16152712_

Round 1

Reviewer 1 Report

Comments and Suggestions for Authors

Your manuscript titled "The Health Impacts of Better Access to Axicabtagene Ciloleucel: The Case of Spain" explores the significant benefits of increased access to CAR T-cell therapy for patients with relapsed/refractory diffuse large B-cell lymphoma. The study's results are compelling. The manuscript is well-written, and the data are presented clearly.

Reviewer 2 Report

Comments and Suggestions for Authors

In this current article “The Health Impacts of Better Access to Axicabtagene 2 Ciloleucel: The Case of Spain” authors believe that the access to axi-cel can lead to potential health benefits that that could be achieved by R/R  DLBCL patients treated earlier with two or more lines of systemic therapy. The study lacks few of the connecting links between the study which needs to be filled in order the study for getting published and authors need to reframe few of the sentences for better representing this article.

1) Authors need to address more about the ZUMA and SCHOLAR trials and their success or failure in context to the present study.

2)Authors should tell the mode of action of Axi-cel in more detail.

3)Fig.1 doesn’t infer what authors mean to say and the legends for the Fig.1 are not explanatory.

4)Are authors sure about the survival events in Fig. 2 are for 500 months instead of 50 months??

5) In Figure 3 authors mention 120 months which is different from the months mentioned in their study, the study lacks the uniformity when mentioned in case of months. The duration of the survival events plays a prominent role.

6)Grammatical errors and spelling should be corrected.

7)Authors should discuss the relevance of their study and how it can be used in future.The novelty of the study should be mentioned.

8) Conclusion lacks the luster and should include robust facts about the study

Reviewer 3 Report

Comments and Suggestions for Authors

1. What is the main question addressed by the research?

This particular research project addresses that he objective of this study was to assess the clinical value of better access to axi-cel in patients with R/R DLBCL who previously received 2 or more lines of treatment in Spain.

2. What parts do you consider original or relevant for the field? What specific gap in the field does the paper address?

This study is done by comparing outcomes with the actual number of patients treated reported in the Spanish National Health System Plan for Advanced Therapies” to the outcomes that would be attained if the full number of patients estimated to be eligible for CAR -T cell therapy for this indication were treated, based on epidemiological estimates.

3. What does it add to the subject area compared with other published material?

Authors describe estimation of the impact of inefficient access to CAR T-cell therapies on the health of the Spanish population in Spain.

4. What specific improvements should the authors consider regarding the methodology? What further controls should be considered?

Authors discuss design details of the the survival benefits associated with axi-cel and considering that 208 patients were treated with CAR T-cell therapy, axi-cel generated an increase of 162% in  LYGs (1,341 LYGs) and 165% in QALYs (1,053 QALYs) compared with chemotherapy and data is in Table 1.

5. Please describe how the conclusions are or are not consistent with the evidence and arguments presented. Please also indicate if all main questions posed were addressed and by which specific experiments.

This study describes estimation of the impact of inefficient access to CAR T-cell therapies on the health of the Spanish population in Spain.

Also provide additional evidence for treatment with CAR T-cell therapies by highlighting the improvements in survival and quality of life that could be achieved with better patient access to treatment.

Authors discuss barriers for delay of CAR T-cell therapies in Spain, due to complex multistep administrative process to the Ministry of Health, late referral of high-risk patients, and the distance from the referring centers to the qualified centers.

Authors discuss potential future actions and strategies to address the issues: these can be simplified.

6. Are the references appropriate?

Yes, references are appropriate.

7. Please include any additional comments on the tables and figures and quality of the data.

There are complex sentences here, the flow gets better by making the sentences simple.

Comments on the Quality of English Language

Please also indicate if all main questions posed were addressed and by which specific experiments.

Authors discuss barriers for delay of CAR T-cell therapies in Spain, due to complex multistep administrative process to the Ministry of Health, late referral of high-risk patients, and the distance from the referring centers to the qualified centers.

Round 2

Reviewer 2 Report

Comments and Suggestions for Authors

All the questions raised by the reviewers have been smartly and efficiently replied by the authors and thus the publication can be accepted.